# Discovery and Genomic Analysis of Three Novel Viruses in the Order *Mononegavirales* in Leafhoppers

**DOI:** 10.3390/v16081321

**Published:** 2024-08-19

**Authors:** Jiajing Xiao, Binghua Nie, Meng-En Chen, Danfeng Ge, Renyi Liu

**Affiliations:** 1Center for Agroforestry Mega Data Science, Haixia Institute of Science and Technology, Fujian Agriculture and Forestry University, Fuzhou 350002, China; jjxiao@fafu.edu.cn (J.X.); lin2454860519@163.com (B.N.); chen_sir_666@163.com (M.-E.C.); 2College of Life Science, Fujian Agriculture and Forestry University, Fuzhou 350002, China

**Keywords:** leafhopper, deep sequencing, *Mononegavirales*

## Abstract

Leafhoppers are economically important pests and may serve as vectors for pathogenic viruses that cause substantial crop damage. In this study, using deep transcriptome sequencing, we identified three novel viruses within the order *Mononegavirales*, including two viruses belonging to the family *Rhabdoviridae* and one to the family *Lispiviridae*. The complete genome sequences were obtained via the rapid amplification of cDNA ends and tentatively named Recilia dorsalis rhabdovirus 1 (RdRV1, 14,251 nucleotides, nt), Nephotettix virescens rhabdovirus 1 (NvRV1, 13,726 nt), and Nephotettix virescens lispivirus 1 (NvLV1, 14,055 nt). The results of a phylogenetic analysis and sequence identity comparison suggest that RdRV1 and NvRV1 represent novel species within the family *Rhabdoviridae*, while NvLV1 is a new virus belonging to the family *Lispiviridae*. As negative-sense single-strand RNA viruses, RdRV1 and NvRV1 contain the conserved transcription termination signal and intergenic trinucleotides in the non-transcribed region. Intergenomic sequence and transcriptome profile analyses suggested that all these genes were co-transcriptionally expressed in these viral genomes, facilitated by specific intergenic trinucleotides and putative transcription initiation sequences.

## 1. Introduction

The identification and characterization of viruses are a pivotal facet within the realm of virology. Traditional detection methods require an evaluation of local phenotypic properties with multiple criteria, including virion morphology, nucleic acid sequences, host range, and pathogenicity. The application of high-throughput sequencing (HTS) and metagenomic data relies on the genome-independent amplification of nucleic acids. This approach allows the simultaneous detection of rare and novel viruses from samples [1,2]. The identification and quantification of novel viruses from HTS have dramatically expanded our knowledge of viruses’ biological processes [3,4]. Numerous novel viruses have been identified and characterized not only in hosts with symptoms but also in symptomless plants or animals [5]. 

Leafhoppers (Hemiptera: Cicadellidae) are pests of many economically important crops, including rice and maize [6]. Mechanical damage caused by leafhoppers during the piercing–sucking process leads to premature leaf abscission and severe yield loss. Additionally, insect vectors, especially a few hemipteran insects, are responsible for the transmission of the majority of plant viruses [7,8,9,10]. *Recilia dorsalis* (Hemiptera: Cicadellidae) is a well-known vector for rice gall dwarf virus (RGDV), rice stripe mosaic virus (RSMV), and other iflaviruses [11,12,13,14,15]. *Nephotettix cincticeps* (Hemiptera: Cicadellidae) is an insect vector for rice dwarf virus (RDV), rice gall dwarf virus (RGDV), rice transitory yellowing virus (RTYV), rice tungro spherical virus (RTSV), and rice tungro bacilliform virus (RTBV). The presence of leafhopper populations causes severe damage to rice and other plants, resulting in yield and quality losses [16,17,18]. The carriage of other viruses can also affect the biological progress of the host through virus–virus interaction [15,19]. Hence, it is important to systematically gather the viral details carried by these insect vectors, which is of paramount importance to avert the harm caused by these vectors. 

Rhabdoviruses contain 434 species of negative-sense single-strand viruses in 56 genera [20,21,22]. Typically, rhabdoviruses consist of five canonical conserved structural proteins with diverse organizations of accessory genes. Their genomes always harbor genes arranged from 3′ to 5′ that encode the nucleocapsid protein (N), phosphoprotein (P), matrix protein (M), glycoprotein (G), and the large multi-functional RNA-dependent RNA polymerase (L). In the canonical genome, the gene junction includes a gene transcription termination polyadenylation (TTP) signal, a non-transcribed intergenic dinucleotide, and a transcription initiation (TI) pentanucleotide sequence. These cis-acting signals control a stop–start (‘stuttering’) sequential gene transcription process, causing attenuation of gene expression based on the distance from the 3′ end [21]. The rhabdoviruses have been described as a threat to a diverse range of organisms, including humans, other vertebrates, invertebrates, and plants. Aphids (Aphididae), leafhoppers (Cicadellidae), and delphacid planthoppers (Delphacidae) are well-known vectors for rhabdoviruses [23]. In nature, most plant rhabdoviruses are transmitted by one or a few related vector species. Unable to be transmitted via plant seeds, rhabdoviruses spread through insect vectors during insect feeding and replicate in these vectors with a persistent-propagative mode [7,24].

The newly established family *Lispiviridae*, belonging to the order *Mononegavirales*, contains a single-stranded negative-sense RNA genome [25,26]. To date, 25 genera and 34 species have been included in the family *Lispiviridae* according to the International Committee on the Taxonomy of Viruses (ICTV). The genome of members of the order *Mononegavirales* consists of core protein genes, i.e., envelope protein genes, and RNA-dependent RNA polymerase genes arranged from the 3′ UTR to the 5′ UTR. Using the transcriptomes of 1243 insects, three arlivirus lineage viruses were discovered, including the Hemipteran arli-related viruses OKIAV94, OKIAV98, and OKIAV99, in greenhouse whiteflies [27]. Recently, another novel arlivirus, Nbu stink bug virus 1 (NbuSBV-1), was identified and characterized from an individual yellow spotted stink bug, Erthesina fullo (Hemiptera: Pentatomidae) [19].

In this study, we investigated the viromes of four leafhopper samples, *Recilia dorsalis* and *Nephotettix virescens*, collected in a rice field, and putative viral sequences were identified from deep-sequencing data. The complete genomes of the three novel viruses were assembled and characterized and tentatively named Recilia dorsalis rhabdovirus 1 (RdRV1), Nephotettix virescens rhabdovirus 1 (NvRV1), and Nephotettix virescens lispivirus 1 (NvLV1). Whole-genome sequences of these viruses were obtained. Sequence alignments and phylogenetic analysis suggested that two are new members belonging to the family *Rhabdoviridae* and one is a novel member of the family *Lispiviridae*. 

## 2. Materials and Methods

### 2.1. Sample Preparation and RNA Extraction

Rice host insects were collected from rice fields in Yunxiao, Zhangzhou Province, and Luoding, Guangdong Province, China (Appendix A), from 2020 to 2021. Three to five insects of the same species were mixed into a single pooled sample for transcriptome sequencing. Total RNA was extracted using TRIzol Reagent (Invitrogen, Carlsbad, CA, USA) according to the manufacturer’s instructions. After the depletion of ribosomal RNA, a cDNA-sequencing library was constructed and sequenced using the MGISEQ-2000 platform to generate 100 bp paired-end reads.

### 2.2. Bioinformatic Analysis 

After obtaining the raw sequencing reads, fastp (version 0.23.2) was used with default parameters to filter adapter sequences and remove low-quality reads (with mean quality scores < 20 in the sliding window of 5 bp from both sides and a length of clean reads less than 50 bp) [28]. The retained clean paired reads were de novo assembled into contigs using Trinity (version 2.1.1) with default parameters [29]. To identify virus-derived sequences, contigs were screened via local BLASTX against NCBI viral protein database (information was downloaded on 2 February 2023, with significant e-values < 1e−5). Contigs longer than 200 nt with hits against virus proteins were kept for further virus identification. Local BLASTN against NCBI nt database helped to remove host-related contigs. Only sequences with optimal hits for viral proteins and no hits for the host animal genomes were treated as potential viral sequences. 

### 2.3. Obtaining the Full-Length Virus Genomes through RACE and Validation of Virus Genome

To obtain the full-genome sequence of viruses, the 3′ and 5′ end sequences of all viral genomes were obtained through rapid amplification of cDNA ends assays using the SMART™ RACE cDNA Amplification Kit (TaKaRa, Dalian, China) according to the manufacturer’s instructions. PCRs were performed using MAX DNA polymerase (TaKaRa, Dalian, China) with the temperature profiles depending on expected product size: 98 °C for 33 s, followed by 34 cycles of 98 °C for 10 s, 55 °C for 30 s, 72 °C 1 kb/10s, and a final elongation step at 72 °C for 10 min. Products were analyzed via electrophoresis on agarose gels. Colonies containing an insert were selected, and plasmid DNA was purified for sequencing. The complete genome sequences were confirmed via Sanger sequencing of overlapping clones (the primers are listed in Appendix A).

### 2.4. Sequence and Phylogenetic Analyses

ORFs of viruses were predicted using ORFfinder with a minimum ORF length of 100 amino acids (aa) using the standard genetic code and other default parameters. The conserved protein domains were predicted using NCBI Conserved Domain Database (https://www.ncbi.nlm.nih.gov/Structure/cdd/wrpsb.cgi, accessed on 23 April 2023). Multiple alignments of RdRp amino acid sequences of these viruses were generated using MAFFT (v7.310) with default parameters [30], and a maximum likelihood phylogenetic tree was constructed using PhyML (version 3.2.0) with 1000 bootstrap replicates [31], with the best substitution model estimated by Prottest3 (version 3.4.2) [32]. To elucidate the classification of candidate viruses from family *Rhabdoviridae*, two viruses, Cultervirus hemicultri and Carbovirus tapeti, were chosen as the outgroup. To elucidate the classification of candidate viruses from family *Lispiviridae*, Orthobornavirus bornaense from the family *Bornaviridae*, and Nyavirus nyamaniniense and Beihai rhabdo-like virus 6 from the family *Nyamiviridae* were chosen as the outgroup.

## 3. Results

### 3.1. Transcriptome Assembly and Virus Discovery after Sample Collection

*Recilia dorsalis* and *Nephotettix virescens* are prevalent insect species within rice fields, which is of significant importance in the context of agricultural virology. To characterize the viromes of these rice host insects, *Recilia dorsalis* and *Nephotettix virescens* specimens were collected from the rice fields for deep-transcriptomic sequencing. An average of 10.96 Gb of data, including 5.85–9.10 × 10^7^ paired-end reads, was generated (Appendix A). A total of 285,936–445,374 contigs were obtained with de novo assembly. After annotating against the local viral-RefSeq database and nt database, 170–280 high-confidence contigs were screened as viral-derived sequences, ranging from 24 to 30 predicted viral families and other unknown groups of viruses (Figure 1).

We found that RD_LD20201212 contains six contigs that shared 97.79–99.50% identity with the proteins of RGDV, which was not detected in RD_LD20200821, the same insect species previously collected in the sample rice field. RD_LD20201212 also contains a contig (9604 nt) originating from rice stripe mosaic virus (RSMV) and another contig that shared 20.91% aa identity with the major core capsid protein of rice ragged stunt virus (RSSV). A contig (4581 nt) in RD_LD20200821 shared 29.19% aa identity with a replication-associated protein of rice stripe necrosis virus (RSNV). These results indicate the diversity of rice viruses carried by the same species at different times. Furthermore, a contig of 8,170 nt in RD_LD20200821 shared 88.06% nt sequence identity with the complete genome of Hangzhou totivirus 12 isolate DGYCFY160 (HzTV12), which was detected in *Recilia dorsalis* from Hangzhou. The predicted ORFs share over 90% nt sequence identity with the viral proteins of HzTV12, suggesting that they are the same virus. In RD_LD20201212, two more iflavirus-like contigs showed high similarity (over 87%) with two reported iflaviruses, Hangzhou recilia dorsalis iflavirus 1 and Hangzhou recilia dorsalis iflavirus 2. The same viruses identified in insect vectors collected from distant regions suggested a distant transmission of insect vectors. In *Nephotettix virescens* (NV_LD20210330), a contig of 10,574 nt contained an ORF that showed 98.00% nt identity (coverage 100%) with the polyprotein of Congyang nephotettix cincticeps iflavirus 1. The complete genome sequence was obtained with RACE (deposited in GenBank under accession number OP614936) and tentatively named Nephotettix virescens iflavirus 1 (NvIV1). Genomic sequence analysis suggested NvIV1 and Congyang nephotettix cincticeps iflavirus 1 are strains of the same viral species within a single family. The systemic analysis of leafhoppers’ viromes helped reveal the high diversity of potential viruses carried by insect vectors.

### 3.2. Characterization of Two Novel Viruses of the Family Rhabdoviridae

According to the obtained viral sequences, two contigs with lengths of 13,704 nt and 14,221 nt shared the highest amino acid identity (33.24% and 33.16%) with the polyprotein of Sanxia Water strider virus 5, suggesting the existence of two viruses belonging to the family *Rhabdoviridae* carried by *Recilia dorsalis* and *Nephotettix virescens*. The full-genome sequences of two viruses were successfully acquired via 5′- and 3′- RACE and validated using Sanger sequencing; they were named Recilia dorsalis rhabdovirus 1 (RdRV1) and Nephotettix virescens rhabdovirus virus 1 (NvRV1), respectively. Without polyA tails, the genome of NvRV1 (deposited in GenBank under accession number OP614936) is 13,726 nt in length, and the full length of RdRV1 (deposited in GenBank under accession number OP614934) is 14,251 nt (Figure 2a).

Both genomes possess a typical genome organization, with five predicted ORFs, a 5′untranslated region (UTR), and a 3′UTR (Figure 2a). According to ORF prediction and CDD analysis, the longest ORF of NvRV1 encodes the putative L protein consisting of 2312 aa, featuring three conserved domains, including an RNA-dependent RNA polymerase (RdRp) domain (1.10e−123), an mRNA-capping region V domain (2.57e−33), and an mRNA-capping enzyme domain (6.22e−33). The 2255 aa L protein of RdRV1 contains another unknown functional domain (DUF1599, 6.17e−03). The A-F motifs as well as the essential GHP (Gly-His-Pro) motif of L protein are conserved in NvRV1 and RdRV1 (Appendix A). The putative N proteins of NvRV1 and RdRV1 are 446 aa and 473 aa. The predicted P protein is 602 aa and 608 aa, followed by an ORF encoding a putative M protein consisting of 245 aa and 256 aa in NvRV1 and RdRV1. ORF4 encodes a putative G protein consisting of 583 and 582 aa. BLASTP analysis conducted in comparison to the online protein database of NCBI suggested that the L protein of NvRV1 is closely related to that of Hangzhou nephotettix cincticeps rhabdovirus 1, with a 35.99% aa identity (coverage: 92%). The NvRV1 N protein is closely related to the N protein of Hemipteran rhabdo-related virus OKIAV26, with a 31.51% aa identity (coverage: 84%).

In the non-coding regions of two viruses, conserved sequences (3′TTAGAAAAA5′) downstream of the stop codon for each ORF were identified as polyadenylation signals for TTP, following the conserved intergenic trinucleotides (Figure 2b). The TTP signal is similar to those of members of betapaprhaviruse [33]. No other transcription initiation sequences (TI) were found between the two ORFs in the genomes of NvRV1 or RdRV1, suggesting that all five genes could be transcribed conjointly. The genome of NvRV1 contains a 168 nt 3′UTR and a 185 nt 5′UTR, while RdRV1 includes a shorter 74 nt 3′UTR and a 154 nt 5′UTR. And the 168 nt leader region at the 3′ terminus of NvRV1 lacks the U7 sequences [21]. The leader region could act as a promoter for RNA synthetic processes, and the full-length complementary sequence was further dissociated to generate RNA template with the help of TTP signal and intergenic trinucleotides. We found that the 11 terminal-most nucleotides showed 90.91% complementarity, with a gap present in only the NvRV1 genome, while longer complementarity sequences (22 nt) were observed in RdRV1 (Figure 2c). The abundance and coverage of NvRV1 and RdRV1 were evaluated using a realignment of the RNA-seq reads to the reconstructed full genome sequence. A total of 154,214 paired-end reads were perfectly mapped to the NvRV1 genome, amounting to 0.11% of all the RNA-seq reads (Figure 2d) and covering 99.91% of NvRV1 genome with 2243 mean depths. In total, 99.78% of the RdRV1 genome was covered by 17,751 paired-end reads.

To figure out the detailed classification of NvRV1 and RdRV1, phylogenetic analysis based on the full lengths of L protein sequences was performed with representative members of the family *Rhabdoviridae* based on the ICTV. The evolutionary model was selected for inferring maximum likelihood trees, according to the lowest Bayesian information criterion. The phylogenetic tree suggested NvRV1 and RdRV1 were in the cluster with species belonging to the genera not assigned to a subfamily (Figure 3). Pairwise sequence identities showed low sequence identities (lower than 60%) in both the genomic sequences and aa sequences of L proteins between RdRV1 or NvRV1 and other individual viruses. Based on the general demarcation criterion for establishing a new species in the family *Rhabdoviridae*, which required a minimum amino acid sequence divergence of 10% in the L proteins, our results suggested that NvRV1 and RdRV1 are two novel viruses in the genera that are not assigned to a subfamily in the family *Rhabdoviridae*.

### 3.3. Characterization of a Novel Member of the Family Lispiviridae

Classic viruses belonging to the family *Lispiviridae* contain a single-stranded negative-sense RNA genome. In this project, a contig consisting of 13,824 nt from NC_YX20210611 shared 31.41% identity with the RdRP protein of Sanxia water strider virus 4 (family *Lispiviridae*) in the Viral RefSeq database. The genome sequence of this candidate virus was obtained via RACE. The full-length genome of this virus is 14,055 nt and was tentatively named Nephotettix virescens lispivirus 1 (NvLV1, deposited in Genbank under accession number OP614930). The genome of NvLV1 has a 142 nt 3′ leader region and a 343 nt 5′ trailer region. The negative-sense genome contains six predicted ORFs in the order 3′UTR-VP1-VP2-P-G-VP5-L-5′UTR (Figure 4a), with the 13 nt terminal-most nucleotides of NvLV1 exhibiting 100% reverse complementarity (Appendix A). In the non-transcribed regions, a conserved transcription termination signal (3′TTTATAAAAA5′) and intergenic trinucleotides (3′GAC5′) were found, followed by a putative transcription initiation sequence (3′GAGAG5′) (Appendix A). No signal was detected in the intergenic region between VP5 and the L protein. 

The blast analysis revealed that VP1 (447 aa) and VP2 (666 aa) showed 66.30% and 27.06% aa sequence identity with the hypothetical GP1 and GP2 protein of Hangzhou lispivirus 1. Furthermore, the P1 (138 aa), G (583aa), and VP5 (362 aa) proteins displayed the highest aa identity with the corresponding proteins of Hangzhou lispivirus 1. The putative VP2 protein was predicted to contain a nuclear localization signal (NLS, 299MSVKRKAGKQ308). The longest ORF of NvLV1 encoded a 2149 aa L protein with three conserved domains: Mononeg_RNA_pol super family (cl15638, 1.29e−122), Mononeg_mRNAcap super family (cl16796, 7.62e−44), and paramyx_RNAcap super family (cl44358, 8.64e−35). Blast analysis indicated that no known RdRP shares more than 45% aa identity with NvLV1 RdRP. RNA-seq reads of *Nephotettix virescens* were remapped to the NvLV1 genome (Figure 4b), and a high abundance of NvLV1 transcripts was observed. 

Using the full length of L proteins with NvLV1 and members belonging to the family *Lispiviridae*, a phylogenetic tree was conducted based on the L protein sequences using the maximum-likelihood method with the LG substitution model, a transition model with empirical base frequencies and free rate heterogeneity (Figure 4c). The result revealed that NvLV1 is a member of the family *Lispiviridae* with the closest relationship with Hemipteran arli-related virus OKIAV94, a member of the genus *Rivapovirus*. Furthermore, the L protein of NvLV1 shares aa sequence identity with the other lispviruses’ L proteins ranging from 21.84% to 41.20%. According to species demarcation criteria with less than <85% identical RdRP amino acid sequences based on the ICTV, NvLV1 is a new member of the family *Lispiviridae*.

## 4. Discussion

Traditional molecular methods have been crucial for the discovery of virus-like genomes in virology, whereas the application of HTS has significantly accelerated the identification and quantification of novel viruses. Rice is one of the world’s most important crops, and rice viruses are responsible for significant losses of rice as well as other crops [34]. It is important to gain a comprehensive understanding of the viromes of these rice host insects, which play a key role in rice virus spread. 

In this study, we conducted deep-sequencing analyses to investigate the potential viruses of leafhoppers (*Recilia dorsalis* and *Nephotettix cincticeps*) collected from rice fields (Appendix A). The detailed investigation indicated the complex viromes of all the samples. With over 20% of the predicted viruses detected in all four samples, 20–23% of the viral sequences are present in each leafhopper, indicating the high diversity of the viromes in these insect vectors. Furthermore, we screened and characterized three novel negative-sense single-stranded RNA viruses, RdRV1, NvRV1, and NvLV1, all belonging to the order *Mononegavirales* (Figure 2, Figure 3 and Figure 4). Gene structure analysis showed that RdRV1 and NvRV1 consist of five predicted ORFs (Figure 2), and the conserved L protein contained the seven conserved motifs for RdRP (Appendix A). 

Since the non-transcribed intergenic region acts as a *cis* signal regulating the termination and reinitiation of the transcription of genes [21,35], these newly identified intergenic trinucleotides (3′CTA5′ for NvRV1, 3′CCA5′ for RdRV1) may participate in regulating the transcription in rhabdovirus. The genome of NvRV1 contains a 168 nt 3′UTR and a 185 nt 5′UTR, while RdRV1 includes a shorter 74 nt 3′UTR and a 154 nt 5′UTR. We found that RdRV1 and NvRV1 contain the conserved TTP signal downstream of each ORF [33], with a consensus sequence, 3′-TTAGAAAAACTA-5′, pinpointed upstream of the start codon without other transcription initiation signal sequences. These results indicate the co-transcription mechanism for all genes in their genomes, which is also suggested by the realignment of the viral genome with the transcriptomic data. Phylogenetic analyses of L protein amino acid sequences showed that RdRV1 and NvRV1 formed a branch with species belonging to genera not assigned to a subfamily (Figure 3). In the current ICTV scheme of Rhabdoviridae taxonomy, viruses are assigned to different species with over 10%-15% sequence divergence of N/L/G proteins [22]. Both RdRV1 and NvRV1 shared less than 60% sequence identity for N/L/G proteins with other known rhabdoviruses. These results suggested that RdRV1 and NvRV1 are two distinct novel members belonging to the family *Rhabdoviridae*.

There is limited information on the recently established family *Lispiviridae* [26]. Similar TTP and TI signals are detected in the non-transcribed regions of NvLV1. Interestingly, no gradient gene expression pattern was found for these genes based on our transcriptomic data, suggesting that NvLV1 may employ a transcription mechanism similar to that observed in rhabdoviruses.

## Figures and Tables

**Figure 1 viruses-16-01321-f001:**
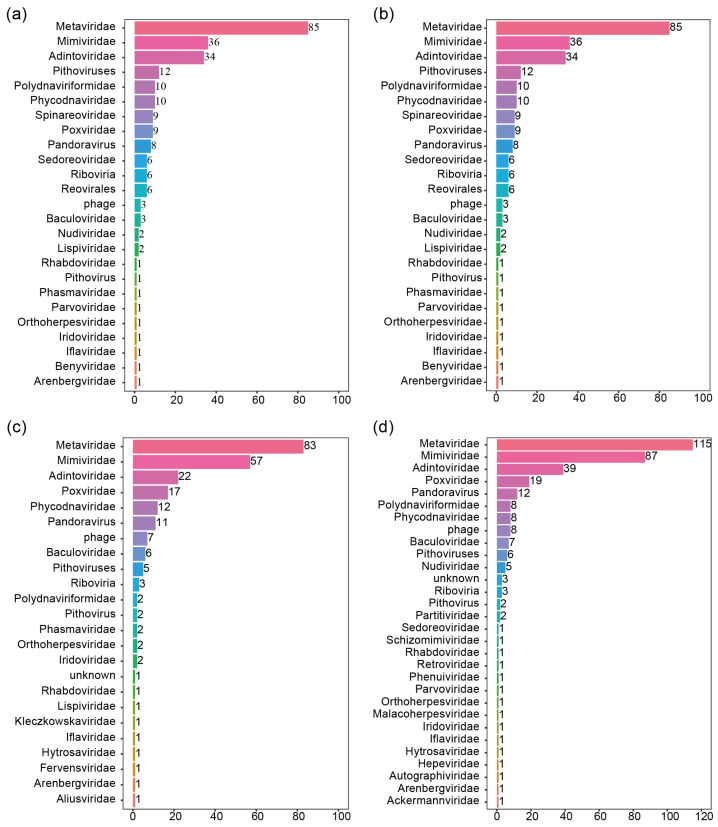
Viromes detected in leafhoppers assigned to different viral families based on the best hits according to BLASTX analysis in RD_LD20200821 (**a**), RD_LD20201212 (**b**), NV_LD20210330 (**c**), and NV_YX20210611 (**d**).

**Figure 2 viruses-16-01321-f002:**
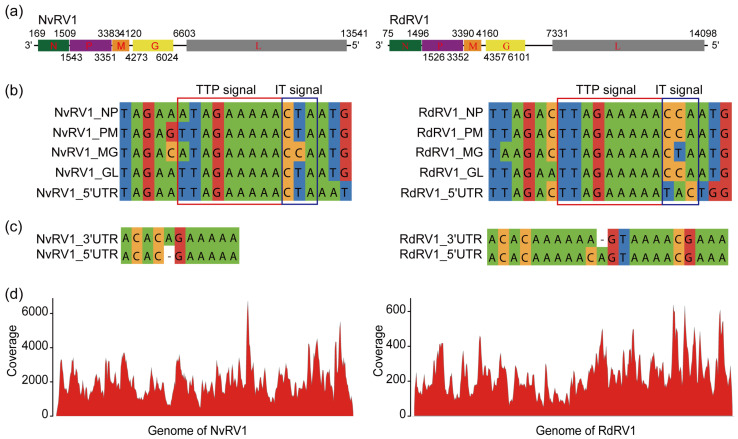
Genome organization of two viruses belonging to the family *Rhabdoviridae*. (**a**) Genome organization of NvRV1 and RdRV1 in the negative sense with the order of 3′UTR-N-P-M-G-L-5′UTR. (**b**) The conserved TTP and IT signals present downstream of the genes in NvRV1 and RdRV1. (**c**) The complementary sequences of most terminal regions in NvRV1 and RdRV1. (**d**) RNA-seq mapping showed fluctuating read distributions in viral genomic RNA.

**Figure 3 viruses-16-01321-f003:**
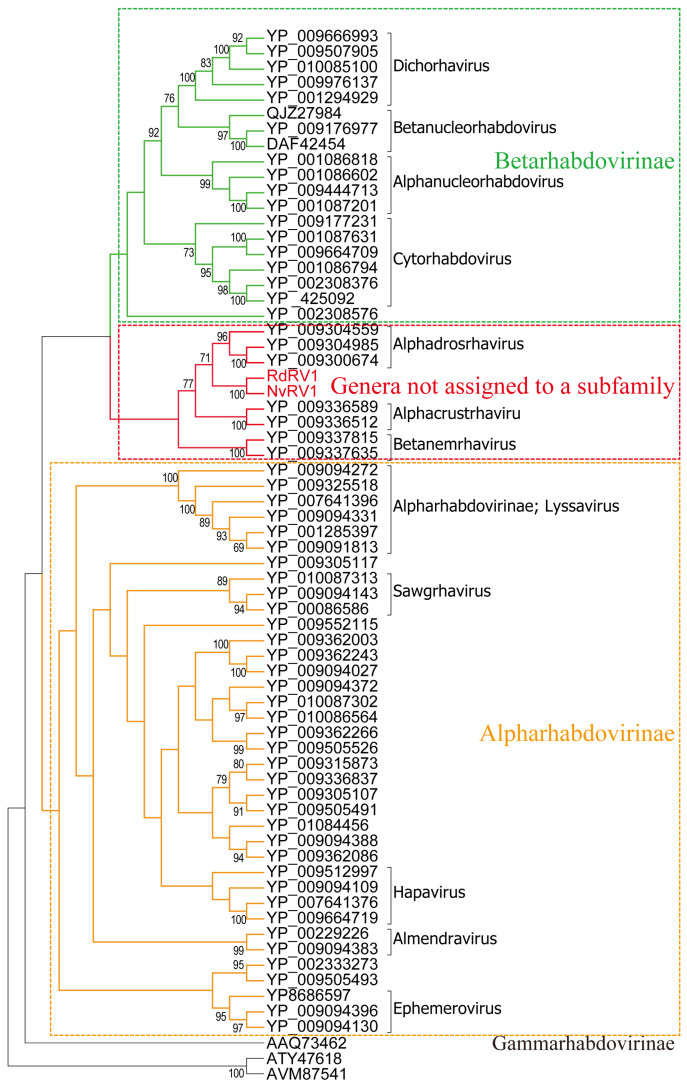
Phylogenetic analysis suggested NvRV1/ RdRV1 are new members of unassigned genera belonging to the family *Rhabdoviridae*. A maximum-likelihood phylogenetic tree based on the amino acid sequence of the full-length L protein of NvRV1/ RdRV1 and other representative members belonging to the family *Rhabdoviridae* was constructed. The corresponding sequence of Novirhabdovirus hirame (AAQ73462), southwest carpet python virus (ATY47618), and Wuhan sharpbelly bornavirus (AVM87541) were used as an outgroup. Bootstrap values (>60%) are shown at each node of the tree. The bar represents the genetic distance.

**Figure 4 viruses-16-01321-f004:**
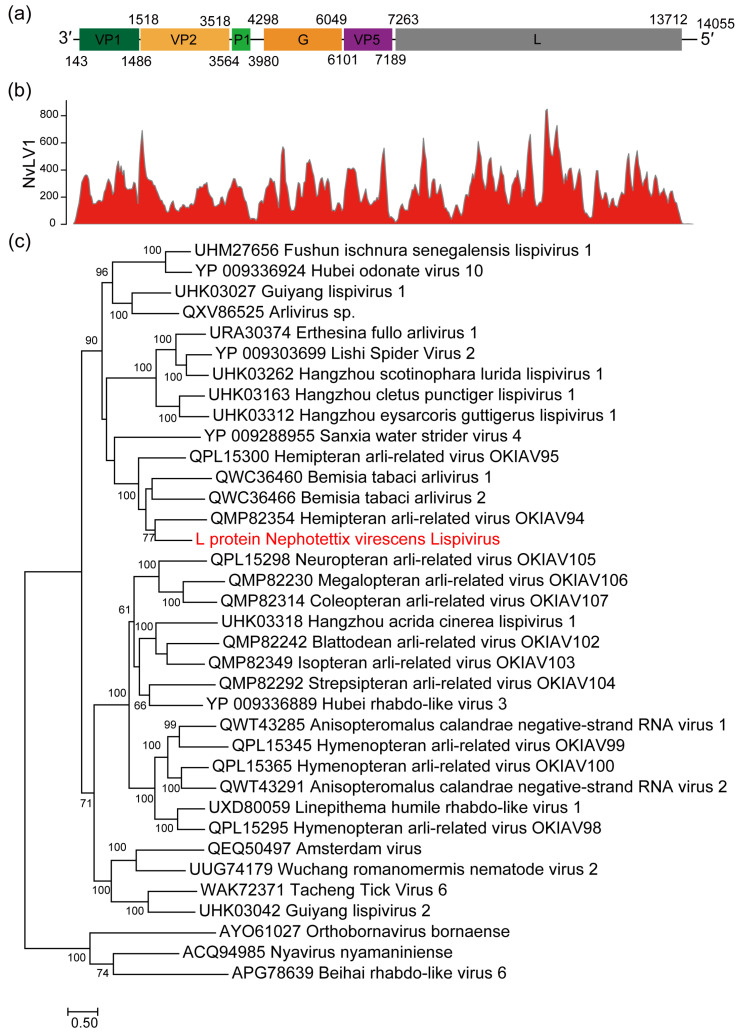
Genomic analysis and phylogenetic analysis suggesting that NvLV1 is a new member in family *Lispiviridae*. (**a**) Genome organization of NvLV1. (**b**) RNA-seq mapping showing abundant read distributions in the NvLV1 genome. (**c**) Maximum-likelihood phylogenetic tree based on the full-length L protein of NvLV1 and other representative members belonging to the family *Lispiviridae*. The corresponding sequences of Orthobornavirus bornaense, Nyavirus nyamaniniense, and Beihai rhabdo-like virus 6 were used as an outgroup. Both trees were generated with PhyML3 using LG+I+G+F substitution model selected via Prottest3 with 1000 bootstraps. Bootstrap values (>60%) are shown at each node of the tree. The bar represents the genetic distance.

## Data Availability

The datasets presented in this study can be found in online repositories. The transcriptomic reads generated in this study have been deposited in the China National Center for Bioinformation of the Beijing Institute of Genomics, Chinese Academy of Sciences (GSA: CRA011034). The full genome sequences have been deposited in the Genome Sequence Archive of the National Genomics Data Center under the identification codes OP614930, OP614934, and OP614936.

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
