# Peer review of "Discovery and Genomic Analysis of Three Novel Viruses in the Order Mononegavirales in Leafhoppers"

_viruses, 2024, doi:10.3390/v16081321_

Round 1

Reviewer 1 Report

Comments and Suggestions for Authors

The manuscript by Jiajing Xiao and colleagues describes the genomic features of three new −ssRNA viruses obtained using analysis of the results of the leafhoppers’ transcriptome sequencing. Since leafhoppers are economically significant pests, studies of their virome are of practical importance, making the manuscript relevant. The figures are good and easy to understand, the manuscript a brief and informative introduction section. The manuscript contains not numerous, but interesting findings and their discussion. The structure of the manuscript is logical. After some revisions it could be considered for publication in the journal “Viruses”.

Line 19 - Please replace “Intergenic” with “intergenomic”.

Line 50 - Please indicate the exact number of genera.

Lines 66-70 - Please rephrase and explain why the family Leviviridae lacks information for species classification.

Line 83 - Please replace “phylogenic” with “phylogenetic”.

Line 105 - Please replace “NT” with “nt”.

Line 109 - Perhaps, “terminal” would be better than “end”.

Line 120 - “ORFs of viruses were predicted using ORFfinder…”

Figure 1. Could you make comments regarding the origin of Mimiviridae, Poxviridae, Phycodnaviridae and Pandoraviridae proteins? Could they indeed belong to cellular proteins?

Figure 3 - I could not find the bar representing genetic distances.

Line 154 and elsewhere - was in amino acid or nucleotide identity?

Line 177 - Please remove “was”

Line 189 - Please replace “consist” with “possess”.

Section 3.2. I would suggest constructing a clustered heatmap of related viruses based in the intergenomic similarity

Line 240 - ‘Maximum-likelihood phylogenetic tree based on the amino acid sequence of the full…’?

Lines 244-245 - “Both trees were generated with PhyML3 using LG+I+G+F substitution model selected by Protest3 with 1000 bootstraps.” - Probably, there is no need to mention PhyML3, the model and how it was found; that should be in the Methods section. Besides, you could just delete the branches with a low bootstrap support instead of removing only the bootstrap values.

Line 257 - should it be “reverse" instead of “inverse”?

Line 263 and elsewhere - please use italics in the names of viral taxa. Figures (a) and (b) do not belong to phylogenetic analysis.

Lines 294-295 - “Rice is one of the world's most important crops, and rice viruses are responsible” - would be better.

Comments on the Quality of English Language

The manuscript needs proofreading.

Author Response

Overall comments:

The manuscript by Jiajing Xiao and colleagues describes the genomic features of three new −ssRNA viruses obtained using analysis of the results of the leafhoppers’ transcriptome sequencing. Since leafhoppers are economically significant pests, studies of their virome are of practical importance, making the manuscript relevant. The figures are good and easy to understand, the manuscript a brief and informative introduction section. The manuscript contains not numerous, but interesting findings and their discussion. The structure of the manuscript is logical. After some revisions it could be considered for publication in the journal “Viruses”.

Response: Thank you for your constructive and detailed comments that help improve the quality of the paper. We have addressed your comments one by one as follows and revised the manuscript accordingly.

Comment 1: (Line 19) - Please replace “Intergenic” with “intergenomic”.

Response: Revised as suggested.

Comment 2: (Line 50) - Please indicate the exact number of genera.

 Response: Thank you for your suggestion. we have updated the species number based on ICTV (https://ictv.global/report/chapter/rhabdoviridae/rhabdoviridae).

Comment 3: (Lines 66-70) - Please rephrase and explain why the family Lispiviridae lacks information for species classification.

Response: Thank you for pointing this out. We examined the description on the International Committee on Taxonomy of Viruses (ICTV) and found the updated species demarcation criteria. We rewrite the sentence and related parts in the result section.

Comment 4: (Line 83)  - Please replace “phylogenic” with “phylogenetic”.

Response: Revised as suggested.

Comment 5: (Line 105)  - Please replace “NT” with “nt”.

Response: Revised as suggested.

Comment 6: (Line 109) - Perhaps, “terminal” would be better than “end”.

Response: The abbreviation RACE stands for Rapid Amplification of cDNA Ends, a molecular biology technique used to determine the unknown sequence at the 5' or 3' ends of a genome. Therefore, keeping the term "ends" is probably better than using "terminal".

Comment 7: (Line 120)  “ORFs of viruses were predicted using ORFfinder…”

Response: Revised as suggested.

Comment 8: Figure 1. Could you make comments regarding the origin of Mimiviridae, Poxviridae, Phycodnaviridae and Pandoraviridae proteins? Could they indeed belong to cellular proteins?

Response: These large DNA viruses are also characterized by their long genome lengths. In our analysis, we found several contigs that showed the highest sequence similarity to viral proteins from members of families Poxviridae, Phycodnaviridae, and Pandoraviridae, suggesting the putative viruses carried by each insect sample. Based on these findings, we described these contigs as highly confident viral sequences.

Comment 9: Figure 3 - I could not find the bar representing genetic distances.

Response: Thank you for pointing out this typo. The bar representing genetic distances was removed to make all species in the right terminal, we have revised the figure with bar and the genetic distances retained.

Comment 10: Line 154 and elsewhere - was in amino acid or nucleotide identity?

Response: the sequence identity is based on amino acid sequences.

Comment 11: Line 177 - Please remove “was”

Response: Revised as suggested.

Comment 12: Line 189 - Please replace “consist” with “possess”.

Response: Revised as suggested.

Comment 13: Section 3.2. I would suggest constructing a clustered heatmap of related viruses based in the intergenomic similarity

Response: Thanks for the suggestion. Constructing a clustered heatmap based on the intergenomic similarity of the related viruses could provide valuable insights into their evolutionary relationships and grouping. The potential function of these intergenomic regions required further analysis based on empirical work.

Comment 14: Line 240 - ‘Maximum-likelihood phylogenetic tree based on the amino acid sequence of the full…’?

Response: Revised as suggested.

Comment 15: Lines 244-245 - “Both trees were generated with PhyML3 using LG+I+G+F substitution model selected by Protest3 with 1000 bootstraps.” - Probably, there is no need to mention PhyML3, the model and how it was found; that should be in the Methods section. Besides, you could just delete the branches with a low bootstrap support instead of removing only the bootstrap values.

Response: Thank you for the suggestion. The two phylogenetic trees were actually generated based on the same substitution model, so our previous description of the models used was somewhat redundant. However, we still believe it is essential to provide the details of the phylogenetic analysis in the legend section for the sake of transparency and to allow readers to fully understand the methodology.

Comment 16: Line 257 - should it be “reverse" instead of “inverse”?

Response: Thank you for pointing this out. Revised as suggested.

Comment 17: Line 263 and elsewhere - please use italics in the names of viral taxa. Figures (a) and (b) do not belong to phylogenetic analysis.

Response: Thank you for the feedback and suggestions. We modify the figure legend from “Phylogenetic analysis” to “Genomic analysis and phylogenetic analysis..” and rewrite the  species name using italics.

Comment 18: Lines 294-295 - “Rice is one of the world's most important crops, and rice viruses are responsible” - would be better.

Response: Revised as suggested.

Reviewer 2 Report

Comments and Suggestions for Authors

Xiao et al describe here the identification of three new virus genomes through deep RNASequencing combined with RACE using template from two leafhoppers. In general, the work is designed and implemented correctly (ie sampling leafhoppers from rice, isolation and sequencing of RNA/cDNA, BLAST analyses and comparisons) and thus confidence can be had that they have identified three new viruses. There were some minor issues associated mostly with communication:

(Lines)

(75-76) Throughout the manuscript, the authors failed to italicize species names such as Erthesina fullo here.

(84-86) The authors state here that they determined characteristics of “prevelance and distribution”, however they both failed to raise this in the abstract AND really failed to address through sufficient sampling. Given the minimal sampling, little can really be inferred from the presence/absence of these and other virus sequences; certainly, prevalence cannot be inferred.

(109) To what does “…sequence of BOTH viral genomes” refer – what is the “both”?

(155-164) Issues wit communication as well as insufficient sampling means as noted above that very limited inferences can be pulled based on their data in regards to prevalence, co-occurrence of viruses more widely in these (and other) leafhoppers, and about virus and insect movement patterns. These types of statements should be removed or very limited. 

(168-170) The sentence “Sequence analysis suggested NvIV1 and Congyang… in one family” makes no sense. Please rewrite to clarify. 

(201-204) Similarity of what – amino acid? Nucleic acid? Specified in other such statements.

(210-212) It is unclear whether there is empirical work here or if the authors are surmising based on their data and the Rodriguez reference clarify.

(216-218) As above authors should clarify if work has been done or it is inferred.

(235-6) Authros should state the criteria for establishing new species here, not just discussion.

(260-1) Authors use double negative “No signal was NOT determined…”, which is confusing – please clarify. 

(300-2) These percent statements don’t make any sense – “20% predicted viruses detected in all four samples, 20-23% viral sequenced are specifically…”. Rewrite to clarify. 

(305) What are “Prevenance analysis” and do you have data?

(331-335) The authors discuss propagation of the insect in light of the viruses – this doesn’t make any sense at all and needs to be completely rewritten to clarify meaning. Further, do you have any evidence that the insect is affected in any way by these viruses to make this section relevant?

Fig 2d, 3d: The authors state in the discussion (279-281) that there is no evidence of transcript gradient in the NvLV1 sample but imply there is one for the other two viruses. The graphs fail to support this. The authors should be more explicit in the data used to make this statement.

Comments on the Quality of English Language

There are numerous instances where the English communication needs significant work, as it likely either muddles or completely changes the authors' intention in their statements. 

Author Response

Overall comments:

Xiao et al describe here the identification of three new virus genomes through deep RNA Sequencing combined with RACE using template from two leafhoppers. In general, the work is designed and implemented correctly (ie sampling leafhoppers from rice, isolation and sequencing of RNA/cDNA, BLAST analyses and comparisons) and thus confidence can be had that they have identified three new viruses. There were some minor issues associated mostly with communication:

Response: Thank you for your thorough review of our manuscript describing the identification of three potentially novel virus genomes from leafhopper samples. We appreciate the constructive feedback, which help us strengthen the presentation and clarity of our findings.

Comment 1:(Lines 75-76) Throughout the manuscript, the authors failed to italicize species names such as Erthesina fullo here.

Response: Thank you for catching this oversight regarding the proper formatting of species names in our manuscript. In the revised version, we carefully review the manuscript and ensure that all species names are properly italicized as per standard scientific convention.

Comment 2:(Lines 84-86) The authors state here that they determined characteristics of “prevelance and distribution”, however they both failed to raise this in the abstract AND really failed to address through sufficient sampling. Given the minimal sampling, little can really be inferred from the presence/absence of these and other virus sequences; certainly, prevalence cannot be inferred.

Response: Thank you for pointing this out. We agree that limited sampling approach does not provide a sufficient basis to make strong conclusions about the prevalence or distribution of these putative viral entities. So, we remove the sentence to predict the prevalence and distribution of our newly identified viruses.

Comment 3:(Line 109) To what does “…sequence of BOTH viral genomes” refer – what is the “both”?

Response: Thank you for pointing this out. The ends of all viruses were obtained through rapid amplification of cDNA ends assays and we revise the relevant section.

Comment 4:(Lines 155-164) Issues with communication as well as insufficient sampling means as noted above that very limited inferences can be pulled based on their data in regards to prevalence, co-occurrence of viruses more widely in these (and other) leafhoppers, and about virus and insect movement patterns. These types of statements should be removed or very limited.

Response: Thank you for pointing this out. I appreciate you highlighting the issues with communication and insufficient sampling that constrain the inferences that can be drawn from the data. You make a fair point that very limited conclusions should be made regarding the prevalence, co-occurrence of viruses, and virus/insect movement patterns based on the current findings. To address this, we significantly limit statements that go beyond what can be reasonably supported by the data.

Comment 5:(Lines 168-170) The sentence “Sequence analysis suggested NvIV1 and Congyang… in one family” makes no sense. Please rewrite to clarify.

Response: we rewrite the relevant sentence as follows:

" Genomic sequence analysis suggested NvIV1 and Congyang nephotettix cincticeps iflavirus 1 are strains of the same viral species within a single family."

Comment 6:(Lines 201-204) Similarity of what – amino acid? Nucleic acid? Specified in other such statements.

Response: Thank you for pointing this out. We have modified the sentences to make it clearer.

Comment 7:(Lines 210-212) It is unclear whether there is empirical work here or if the authors are surmising based on their data and the Rodriguez reference clarify.

Response: We are making inferences based on the available evidence, rather than reporting directly observed results, we will qualify our language accordingly in the discussion section.

Comment 8:(Lines 216-218) As above authors should clarify if work has been done or it is inferred.

Response: As above, we also limited the statements here to avoid unsupported extrapolations.

Comment 9:(Lines 235-6) Authors should state the criteria for establishing new species here, not just discussion.

Response: Thank you for pointing this out. We have added the species demarcation criteria. 

Comment 10:(Lines 260-1) Authors use double negative “No signal was NOT determined…”, which is confusing – please clarify.

Response: We have rewritten the sentence to make it clearer.

Comment 11:(Lines 300-2) These percent statements don’t make any sense – “20% predicted viruses detected in all four samples, 20-23% viral sequenced are specifically…”. Rewrite to clarify.

Response: We agree that the phrasing is confusing and requires revision. To address this, we rewrited the relevant sentences to provide a clearer and more coherent presentation of the data: "The detail investigation indicated the complex virome in all samples. With over 20% of the predicted viruses detected in all four samples, 20%-23% of viral sequenced are specifically present each leafhopper, suggesting the highly diversity of virome in those insect vectors. "

Comment 12:(Lines 305) What are“Prevenance analysis” and do you have data?

Response: Thank you for pointing this out. We have removed sentence since published datasets is not solid for prevenance analysis.

Comment 13:(Lines 331-335) The authors discuss propagation of the insect in light of the viruses – this doesn’t make any sense at all and needs to be completely rewritten to clarify meaning. Further, do you have any evidence that the insect is affected in any way by these viruses to make this section relevant?

Response: Thank you for pointing out the issues with the discussion of insect propagation in relation to the viruses. The sentence about the propagation of the insect in light of the viruses is problematic. In the revised manuscript, we removed statements about the viruses influencing the insect's propagation or population dynamics and focus the discussion solely on the genomic characteristics of the novel viral sequences identified.

Comment 14: Fig 2d, 3d: The authors state in the discussion (279-281) that there is no evidence of transcript gradient in the NvLV1 sample but imply there is one for the other two viruses. The graphs fail to support this. The authors should be more explicit in the data used to make this statement.

Response: The transcript gradient is present in several members of the order Mononegavirales, but is not present in all three of the novel viruses identified in our results.

Reviewer 3 Report

Comments and Suggestions for Authors

Deep sequencing has identified three potentially novel viruses in leafhoppers.  The work presented appears to have been carefully carried out and is presented logically and clearly.  I don't have any concerns about the data presented.  However, deep sequencing will always find evidence of numerous virus genomes in insects.  The question is whether or not it is just the sequences that exist or whether the sequences relate to actual viruses?   What evidence can the authors point to that the sequences they have identified are 'real' viruses?  Have they detected mRNAs, proteins, virus particles?  If not, the authors need to modify their statements accordingly e.g. putative novel viruses have been identified or novel virus genomes have been identified.

Please could the authors also address a point in the introduction (lines 46-47).  It is stated that simply finding evidence of virus sequences is important to 'avert harm' caused by the virus vector.  Please could this be expanded to explain how such data 'averts harm' or remove this sentence.  I can't see how sequence data leads directly to averting harm.

Author Response

Overall comments:

Deep sequencing has identified three potentially novel viruses in leafhoppers.  The work presented appears to have been carefully carried out and is presented logically and clearly.  I don't have any concerns about the data presented.  However, deep sequencing will always find evidence of numerous virus genomes in insects.  The question is whether or not it is just the sequences that exist or whether the sequences relate to actual viruses?   What evidence can the authors point to that the sequences they have identified are 'real' viruses?  Have they detected mRNAs, proteins, virus particles?  If not, the authors need to modify their statements accordingly e.g. putative novel viruses have been identified or novel virus genomes have been identified.

Response: Thank you for the thoughtful feedback on our manuscript. We appreciate you taking the time to provide these constructive comments, which will help us strengthen the presentation and clarity of our findings.

Comment 1: The question is whether or not it is just the sequences that exist or whether the sequences relate to actual viruses? What evidence can the authors point to that the sequences they have identified are 'real' viruses?  Have they detected mRNAs, proteins, virus particles?  If not, the authors need to modify their statements accordingly e.g. putative novel viruses have been identified or novel virus genomes have been identified.

Response: Thank you very much for your suggestion. You raised an important concern about distinguishing between the detection of viral sequences versus the identification of genuine, functional viruses. We acknowledge that deep sequencing alone does not conclusively demonstrate the existence of active viral entities. Based on deep sequence, we indeed found numerous virus genomes. For those three newly identified viruses, we detected the presence of those viruses in insect samples based on PCR reactions, which were presented in the supplementary figures. We will revise our language throughout the manuscript to refer to other viral sequences as "putative" or "candidate" viruses.

Comment 2: Please could the authors also address a point in the introduction (lines 46-47).  It is stated that simply finding evidence of virus sequences is important to 'avert harm' caused by the virus vector.  Please could this be expanded to explain how such data 'averts harm' or remove this sentence.  I can't see how sequence data leads directly to averting harm.

Response: Thank you again for the additional feedback. You are correct that the statement in lines 46-47 about the importance of virus sequence data for "averting harm" caused by the vector requires more explanation or revision. As written, this claim is overly broad and not sufficiently substantiated. In the revised introduction, we remove this particular sentence altogether, as it may be an overreach without providing the necessary supporting context.

Round 2

Reviewer 1 Report

Comments and Suggestions for Authors

Taking into account the responses and clarifications of the authors, there are no longer any significant comments on the article “Discovery and genomic analysis of three novel viruses in the order Mononegavirales in leafhoppers”.